

# 1 Efficient droplet activation of ambient black carbon particles in sub-

# 2 urban environment

Ping Tian[1], Dantong Liu[2*], Kang Hu[2,3], Yangzhou Wu[2,4], Mengyu Huang[1,5], Hui He[1,5], Jiujiang Sheng[1,5],
Chenjie Yu[6], Dawei Hu[7], Deping Ding[1,5]
[1.] Beijing Key Laboratory of Cloud, Precipitation and Atmospheric Water Resources, Beijing Meteorological Service,
Beijing, 100089, China
[2.] Department of Atmospheric Sciences, School of Earth Sciences, Zhejiang University, Hangzhou 310058, China
[3] Jiangsu Collaborative Innovation Center of Atmospheric Environment and Equipment Technology, Jiangsu Key Laboratory
of Atmospheric Environment Monitoring and Pollution Control, Nanjing University of Information Science & Technology,
Nanjing, 210044, China
[4] College of Environmental Science and Engineering, Guilin University of Technology, Guilin, 541006, China
[5] Field Experiment Base of Cloud and Precipitation Research in North China, China Meteorological Administration, Beijing,
100089, China
[6] Université Paris Cité and Univ Paris Est Créteil, CNRS, LISA, Paris, F-75013, France
[7] Department of Earth and Environment Sciences, University of Manchester, Manchester, UK.
*Corresponding to:* Dantong Liu (dantongliu@zju.edu.cn)



**Abstract**
The cloud condensation nuclei (CCN) activity of black carbon particles (BC) importantly determines their
impacts on cloud microphysics and atmospheric lifetime. This process is crucially determined by the amount of
more hygroscopic coating materials BC can acquire during aging process. It remains a challenge for ambient
measurements to capture this process and link this with CCN activity of BC. Here by using coupled
measurements of size-resolved number concentrations of CCN at configured water supersaturation (SS) and BC-
containing particles (BCc), we are able to directly monitor the droplet activation diameter (D50) and activation
fraction of BCc ($F_{act,BC}$) in ambient environment. The number concentration of BCc was found to peak at
diameter 180-210 nm after acquiring coatings, larger than that for all particles (50-150 nm). This led to that the
initially smaller BC can be enlarged and become larger and more hygroscopic, even having higher CCN activities
than other particles. The $F_{act,BC}$ increased from 42% to 69% in number and from 67% to 85% in mass, as SS
increased from 0.1% to 0.2%, but tended to reach a plateau when SS>0.2%.Notably, $F_{act,BC}$ and D50 linearly
correlated with equivalent photochemical age, at a rate of +2% per hour and -3nm per hour, respectively. The
results suggest BCc from anthropogenic sources can readily serve as CCN at a relatively low SS, and over half of
the BC population can be activated in a few hours. This means the surface-sourced BC can efficiently incorporate
into clouds and may exert important indirect radiative impacts.
**Keywords:** black carbon; cloud condensation nuclei; droplet activation fraction; photochemical age



## Introduction

Black carbon (BC), a significant anthropogenic pollutant, generate from the incomplete combustion of fossil fuels and profoundly influences the climate through direct (Ramanathan and Carmichael, 2008; Bond et al., 2013), indirect (Koch et al., 2011), and semi-direct (Koch and Del Genio, 2010) effects. The climate response to BC is sensitive to its lifetime (Hansen et al., 1997), which remains a considerable uncertainty in model studies due to spatial and temporal variations (Park et al., 2005; Koch et al., 2009; Lund et al., 2017).

Wet scavenging, the primary removal mechanism of atmospheric BC (Jacobson, 2010), that BC particles serve as cloud condensation nucleation (CCN) activated into the cloud droplets and removed through precipitation. Although freshly emitted BC particles are generally hydrophobic, they become more hygroscopic through atmospheric processing, acquiring soluble materials like sulfates, nitrates, and secondary organic aerosol compounds, or by coagulating with other particles (Dusek et al., 2006; Tritscher et al., 2011). The increased hygroscopicity enhances CCN activation efficiency of BC (Weingartner et al., 1995), yet there is significant variability in the literature regarding the degree to which BC serves as CCN (Wu et al., 2019; Rose et al., 2011), due to the diversity in properties of BC resulting from its various sources and atmospheric aging process (Henning et al., 2012; Dalirian et al., 2018).

The $\kappa$-Köhler theory (Petters and Kreidenweis, 2007), combined with the ZSR mixing rule (Pruppacher and Klett, 1997; Henning et al., 2010), has been successful in characterizing the hygroscopicity of internally mixed BC, assuming a spherical core-shell structure. Chamber studies where the physic-chemical conditions and the initial emitted particles are well controlled indicate that the hygroscopicity of BC is determined by the fraction and hygroscopicity of the coating material (Snider et al., 2010; Friebel et al., 2019). A relatively thin soluble coating like 2nm corresponding to about 5% of the total volume for 250nm particles are enough to make hydrophobic BC CCN active under typical atmospheric supersaturation (Dalirian et al., 2018). The hygroscopic properties of the coating materials can vary significantly; notably, increase of the hygroscopicity of BC induced by secondary inorganic aerosols (e.g., ammonium nitrate) is more pronounced than that caused by organic aerosols (Kuwata et al., 2007; Liu et al., 2013). However, laboratory experiments often fail to replicate the complexity of atmospheric conditions, comprehensive field measurements are need to capture the real-time evolution of the CCN properties of BC.

Despite the advancements in understanding the physicochemical transformations of BC through field observation (Liu et al., 2017), quantifying its CCN activity in the ambient conditions remains challenging due to the variability in its sources, composition, mixing state, and atmospheric processes (Zhang et al., 2008). Direct field observations indicate that number fraction of 50%–90% BC can be activated into cloud or fog, dependent on actual ambient supersaturation (Motos et al., 2019a; Motos et al., 2019b;). More than 50% of BC mass can been removed through the wetting removing process during the transport from surface to planetary boundary layer (PBL), as evidenced in studies in East Asia polluted area (Kondo et al., 2016; Liu et al., 2020) in east Asian polluted area, but it's difficult to know true ambient supersaturation in the in-situ cloud observation. However, true ambient supersaturation is challenging to determine in in-situ cloud observations.

The hygroscopic of BC particles can be derived from the measured mixed state of BC by size resolved soot particle aerosol mass spectrometer (SP-AMS) (Onasch et al., 2012; Wu et al., 2019), and the hygroscopicity tandem differential mobility (HTDMA) instrument under sub saturation condition, however, only limited atmospheric CCN efficiency studies of BC particles were conducted under the water supersaturation condition. Continuous-flow cloud condensation nuclei counter (CCNC) can measure the CCN number concentration of the bulk aerosol under well-control supersaturation condition (Roberts and Nenes, 2005), but it cannot distinct how many parts of BC are activated as the number proportion of BC account for about 10% of the ambient aerosol. Recently, Hu et al. (2021a) developed an novel method to direct observe the number activation ratios of BC





under certain supersaturation, based on the size resolved CCNC and BC measurements, revealing potential
overestimation of hygroscopicity of BC based on bulk chemical composition.
In this study, continuous observation of the CCN activation and hygroscopic properties of BC under water
supersaturation (0.1%, 0.2%, and 0.3%) were conducted during springtime in a sub-urban site in Beijing. The
variation of the hygroscopicity of BC during different pollution levels was investigated according to classification
of four different air masses clusters. The equivalent photochemical age ($t_{age}$), calculated based on the Proton-
Transfer-Reactor Time-of-Flight Mass Spectrometer, was used to characterize the evolution of the CCN
activation properties of BC under ambient atmospheric aging.

## 2. EXPERIMENTAL AND METHODS

### 2.1. Measurements site and Instruments Setup

The filed campaign was conducted for more than 1 month from 29th Apr. to 2nd Jun. 2020 at Beijing Cloud
Laboratory and Observational Utilities Deployment Base (CLOUD Base) (Tian et al., 2022), which is
approximately 65Km to the northeast of the central Beijing, with the main local emissions are from industrial,
traffic, and residential, which represent a typical sub-urban environment. Higher pollution condition at this site
was correlate with pollutant regional transport from the west and south pollutant area, while clean condition was
attribute to the pollutant dilution by the northwest clean air (Hu et al., 2022).
The instruments setup was shown in Figure 1a. The bulk and size-resolved physic-chemical and hygroscopicity
properties of aerosol were simultaneously measured during the field campaign. A cyclone PM$_{2.5}$ cut-off inlet
(Model: SCC1.829, BGI Inc., USA) was used to remove the coarse mode particles (larger than 2.5μm), and the
air was dried by a Nafion tube before entering the instruments. In addition to particle measurement, volatile
organic compounds (VOCs) were measured by a Proton-Transfer-Reactor Time-of-Flight Mass Spectrometer
(PTR-TOF-MS 8000, Ionicon Analytik GmbH Innsbruck, Austria) (Fig. 1a).

### 2.2 Aerosol Chemical Components Measurements and Hygroscopic Growth Factor Calculation

The mass concentration of non-refractory submicron aerosol species including organic aerosol (OA), chloride
(Chl$^-$), ammonium (NH$_4^+$), nitrate (NO$_3^-$) and sulfate (SO$_4^{2-}$) was measured by the Aerodyne high-resolution time-
of-flight AMS (HR-ToF-AMS, Aerodyne Research Inc., USA). The principle and instruction of HR-ToF-AMS
was already detailed in previous publications (Jayne et al., 2000; Drewnick et al., 2005). The HR-ToF-AMS was
calibrated before and after the experiment, and relative ionization efficiencies of NH$_4^+$ (4.0) and SO$_4^{2-}$ (1.2) were
calibrated using pure ammonium nitrate and ammonium sulfate, respectively, while default relative ionization
efficiency values were used for OA (1.4), NO$_3^-$ (1.1), and Chl$^-$ (1.3) (Middlebrook et al., 2012).
The measurement of individual refractory black carbon (rBC)-containing particles (BCc) and BC-free particles
(BCf) was conducted using the Single Particle Soot Photometer (SP2, DMT Inc., USA). The SP2 employs an
intense $\lambda$=1064 nm Nd:YAG laser to heat BCc until incandescence, with the mass and diameter (Dc) of rBC
quantified based on the intensity of the incandescence signal. Prior to the experiment, the incandescence signal
was calibrated using Aquadag standards (Acheson Inc., USA) and was corrected for ambient BC using a factor of
0.75 (Laborde et al., 2012).
The measurement of individual refractory BC (rBC)-containing particles (BCc) and BC-free particles (BCf) was
conducted using the Single Particle Soot Photometer (SP2, DMT Inc., USA). The SP2 use intense 1064nm
Nd:YAG laser beam to heat BCc to vaporize, with the mass and diameter ($Dc$) of rBC quantified based on the
intensity of the incandescence signal. The incandescence signal was calibrated using Aquadag standard (Acheson
Inc. USA) and was corrected for ambient BC using a factor of 0.75 before the experiment (Laborde et al., 2012).
Particles exhibiting only scattering signal were classified as BCf, with their diameters determined based on
intensity of scattering signal, calibrated by polystyrene latex spheres (PSLs). Due to the distortion of the



scattering signal caused by the vaporization of the coating material of BCc, the leading edge only (LEO) fitting
method was employed to derive the entire size of BCc (Gao et al., 2007; Liu et al., 2014). As the optical diameter
of BCc is largely independent of the morphology of BCc, the measured optical diameter by SP2 is assumed to be
equal to the volume equivalent diameter ($D_{ve}$) (Moteki et al., 2010; Hu et al., 2021b). Consequently, the relative
bulk volume of coating to BC ($V_{coating}/V_{BC}$) was calculated as the cubic ratio of diameter of $D_{ve}$ and $D$c (Liu et al.,
128 2017):

$$VR = \frac{V_{coating}}{V_{BC}} = \frac{\sum_i D_{ve}^3}{\sum_i D_c^3} - 1 \qquad (1)$$

where $D_{ve}$ is volume equivalent diameter of the entire BCc, and $D_c$ is the rBC core diameter for the $i^{th}$ single BCc ,
respectively.
The hygroscopic growth factor ($\kappa$) which depend on the chemical properties (i.e. molecular weight, dissociate,
and osmotic coefficient, etc.) can be used to predict the activation properties of particles as CCN under certain
supersaturation (SS) (Petters and Kreidenweis, 2007). The hygroscopicity growth factor ($\kappa$) for all-particles ($\kappa_{all}$)
were calculated applying Zdanovskii-Stokes-Robinson (ZRS) mixing rule for the volume contribution from
different chemical compositions(Mcdonald, 1953). The expression for $\kappa_{all}$ is:

$$\kappa_{all} = \sum_i \varepsilon_i \, \kappa_i \qquad (2)$$

where $\kappa_i$ and $\varepsilon_i$ represents the hygroscopic growth factor and volume fraction for each chemical component (i.e.
$(NH_4)_2SO4$, $NH_4HSO_4$, $NH_4NO3$, and BC etc.).
The $\kappa$ of single BCc ($\kappa_{BC}$) is depended on the volume fraction and $\kappa$ of the coating material, and calculated as:

$$\kappa_{BCc} = \kappa_{coating} \times (\frac{VR}{1+VR}) \qquad (3)$$

where the $\kappa_{coating}$ was chosen as the $\kappa_{all}$ which also represent the average $\kappa$ of particles, and $V_{coating}/V_{BC}$ is
volume ratios of coating material over the BC, which reflect the coating thickness of BCc particles. The $\kappa_{BC}$ was
test successfully to predict the activation of BCc as CCN in the field in-situ cloud observation (Motos et al.,
2019a).
**2.3 Size Resolved CCN Activity of All-particles and BCc**
Atmospheric polydisperse aerosols were classified into monodisperse particles at the outlet of a Differential
Mobility Analyser (DMA, Model: 3081, TSI Inc., USA). Particles of different mobility sizes were selected by
applying varying voltage to the DMA, maintaining a specific sheath flow rate (Knutson and Whitby, 1975).
Instead of employing the standard TSI Aerosol Instrument Manager control software, the DMA was governed by
custom-made software, enabling the selection of specific diameters and the control of residence time at those
diameters (Moore et al., 2010; Deng et al., 2011). A total of 29 mobility diameters ranging from 29 nm to 612 nm,
at logarithmic intervals, were chosen, and each diameter was maintained for 30 seconds. This time was selected
in consideration of the time lag (estimated to be less than 20s) between the instruments following the DMA (Fig.
1a). A complete size scan took approximately 15 minutes. The DMA's sheath and sample flow rates were
calibrated both before and after the experiment, and the mobility size selected by the DMA was validated using
PSLs.
The monodisperse number concentration of all condensation nuclei ([$CN$]), [$CCN$], and both [$BCc$] and [$BCf$] was
measured by a Condensation Particles Counter (CPC, Model 3772, TSI Inc., USA) with a flowrate of 0.30 L/min,
and a Cloud Condensation Nucleus counter (CCNc, DMT Inc., USA) with a flowrate of 0.50 L/min, and the other
Single Particle Soot Photometer (SP2, DMT Inc., USA) with a flowrate of 0.12 L/min, respectively (Fig. 1a).



Three SSs at 0.1%, 0.2%, 0.3% were set for the CCNc, and with an interval of 30 minutes, allowing two size
scans for each SS. The sample/sheath flowrate and also the SS of the CCNc were calibrated before and after the
experiment (Rose et al., 2008). Moreover, the monodisperse [CCN], [CN], and [BCc] were corrected for multiple
charge effects, which performed from large particles (only have one negative charge) to smaller particles
according to charging probabilities for different mobility size (Wiedensohler, 1988).
The CCN activity of all-particles was determined based on the size resolved activation ratio ($F_{act}$) ([CCN]/[CN])
follow the method by Moore et al. (2010). This size resolved ratio, [CCN]/[CN], reflect how many parts of CN
could serve as CCN can be directly measured by the CCNc and CPC with size scanning by the DMA. [CCN]/[CN]
increase with the particle size, the ratio equals 1 at larger particle sizes (e.g., greater than 300 nm), whereas it
reduces to 0 at smaller sizes (e.g., 30 nm). The specific size at which the [CCN]/[CN] ratio reaches 0.5 is
identified as the activation diameter (D50). This critical value is generally derived through sigmoid fitting of the
size-resolved [CCN]/[CN] (Rose et al., 2008).
The CCN activated number concentration of BCc [$BC_{act}$] is discriminated from BCf and directly measured in this
study follow the method by Hu et al. (2021a), which compare the difference between the un-activated particle
number concentration ([$CN_{unact}$]) and [BCc] to derived how many BCc are activate into CCN. [$CN_{unact}$] is derived
by subtracting the [CCN] from the total [CN]. [$CN_{unact}$] is zero at larger particle sizes, indicating that all particles,
including BCc, are activated. Conversely, when [$CN_{unact}$] equals the total [CN], it signifies that none of the
particles are activated. It is well recognized activation of BCc into CCN is more challenging compare to that of
BCf at the same diameter, due to BCc's lower $\kappa$ (Dusek et al., 2006; Bond et al., 2013). Only when the [$CN_{unact}$] is
smaller than [BCc], it means some BCc have been activated into CCN, while [$CN_{unact}$] is zero, it means all BCc
are activated. So, the [$BC_{act}$] and activation ratios ($F_{act,BC}$) of BCc are calculated as:

$$[BC_{act}] = [BCc] - ([CN] - [CCN]) \qquad (4)$$

$$F_{act,BC} = ([BC_{act}]/[BCc]) \qquad (5)$$

[$BC_{act}$] is the difference between the [BCc] and [$CN_{unact}$], and the $F_{a,BC}$ is calculated by [$BC_{act}$] over the [BCc].
When [$CN_{unact}$] is larger than [BCc] means none of BC could activate, the [$BC_{act}$] and $F_{act,BC}$ is 0.

**2.4 Determination of the Photochemical Age**

The PTR-TOF-AMS mass concentration measurement was calibrated by the CH3COCH4+ (m/z 59.0490) and
H3O+ (m/z 21.0226) following the procedures described by Bruns et al. (2016). The equivalent photochemical
age ($t_{age}$), reflected the OH exposure dose of the air masses, was calculated as the change of toluene over benzene
concentration ratio due to their different reaction coefficient with OH, and expressed as:

$$t_{age} = \frac{1}{[OH](k_{toluene} - k_{benzene})} \times \left[\ln\left(\frac{[toluene]}{[benzene]}\right)_0 - \ln\left(\frac{[toluene]}{[benzene]}\right)\right] \qquad (6)$$

where [OH] is the mean ambient OH concentration at $2.5 \times 10^6$ molecule cm$^{-3}$ during summer daytime in this
region (Yuan et al., 2012). $k_{toluene}$ and $k_{benzene}$ is the reaction rate with OH of toluene ($5.63 \times 10^{-12}$ cm$^{-3}$
molecule$^{-1}$ s$^{-1}$) and benzene ($1.22 \times 10^{-12}$ cm$^{-3}$ molecule$^{-1}$ s$^{-1}$), respectively. $\left(\frac{[toluene]}{[benzene]}\right)_0$ is the ratio of freshly
emitted toluene and benzene which is assumed to be 3.3, and $\frac{[toluene]}{[benzene]}$ is the ration of measured toluene and
benzene in the atmosphere. The calculation of $t_{age}$ was only performed for the daytime to warrant it indicates the



photochemical process. The dataset of $t_{age}$ in this study was already test successfully to obtain the reaction or
production rates of secondary OA and VOCs (Wu et al., 2022).

**2.5. Backward Trajectory Analysis**

The potential source contribution to the target site was modelled by using Hybrid Single-Particle Lagrangian
Integrated Trajectory Model (HYSPLIT) ensemble calculation with 3-hourly, 1°×1° GDAS horizontal and
vertical reanalysis wind field to retrieve backward 36-hour air mass to the target site, and air mass clusters came
from four different directions were classified during the entire observation periods (Fig. 1b). Cluster 1 (C1) was
from the southern polluted area, while C2-C4 were from the northern clean areas (Fig. 1b).

**3. RESULTS AND DISCUSSION**

**3.1 Overview of the Aerosol Physical and Chemical Properties**

Figure 2a-2d shows the time series of aerosol physical and chemical properties with the colour labels at the top
represent different air masses influence periods, and the statistical comparison of aerosol properties between
different air masses were show in Fig. 2e-2i. Consistent with previous studies, the aerosol physic-chemical
properties were highly dependent on air masses (Tian et al., 2020; Zhao et al., 2020). Higher pollutant
concentrations including $PM_1$, Sulphate, Nitrate, OA, and BC were observed during C1 period with air from
southern part of North China Plain (NCP) (average $PM_1$ value of $33.2 \pm 16.4$ μg m$^{-3}$) (Fig. 2e and Fig. 2f), where
contain high pollutant concentration due to the intensive emissions (Fig. 1b). When the air come from northern
clean part of NCP during C3 and C4 period (Fig. 1b), lower pollutant concentrations were observed (average $PM_1$
value of $10.4 \pm 5.8$ μg m$^{-3}$) (Fig. 2a and Fig. 2e), which suggest pollutants were diluted and dispersed during the
fast transport process. Thus, C1 was estimated as polluted condition, while C3 and C4 was classified as clean
environment. However, C2 came from the same direction as C4 but with less distances of transportation, which
means smaller wind speeds to dilute the emissions, so the $PM_1$ mass concentration ($15.4 \pm 10.6$ μg m$^{-3}$) of C2 was
higher than clean environment, but still one half lower than that of C1, and classified as light pollution condition
(Fig. 2e).
During the transition from clean to pollution condition, the mass concentrations of all chemical components
experienced substantial increase, as shown in Fig. 2a. Notably, the increase in inorganic aerosols (including
$NH_4HSO_4$, $(NH_4)_2SO_4$, and $NH_4NO_3$) was more pronounced than that of OA (Fig. 2b), which lead to lower/higher
contribution of OA to all-particles during polluted/clean period (Fig. 2b and Fig. 2e). As $\kappa_{all}$ is determined based
on the volume fraction of OA and inorganic aerosol according to the ZSR internal mixing rule, this result distinct
$\kappa_{all}$ characteristic for different airmasses influence periods (Fig. 2d and Fig. 2h).

**3.2 The Hygroscopicity of BCc and All-particles**

The $\kappa_{all}$ (particles average hygroscopicity) varied from 0.13 to 0.45 with an average value of 0.29 during
observation (Fig. 2d), the $\kappa_{all}$ during C1 polluted period (average value of 0.38) was 1.5 times larger than that
during C3/C4 clean period (average value 0.25) (Fig. 2h), this was attributed to the increased contribution of
inorganic aerosol, known for their high hygroscopicity, under polluted condition.
The $\kappa_{BC}$ is influenced by a combination of $\kappa_{all}$ and the coating thickness (reflected by $V_{coating}/V_{BC}$) (Eq. 3). The
$V_{coating}/V_{BC}$ of BCc during C1 polluted period was $4.9\pm4.2$, which were about 3 - 5 times larger than that during
C3/C4 clean period (Fig. 3c and Fig. 3g), implying thick coating of BCc under polluted condition. Due to both
increase of $V_{coating}/V_{BC}$ and $\kappa_{all}$, the $\kappa_{BC}$ show similar variation characteristic as $\kappa_{all}$ also with higher value under
polluted condition (average value of 0.23), while lower value under clean condition (average value of 0.11) (Fig.
2d and Fig. 2h).



$V_{coating}/V_{BC}$ exhibits a larger variation, ranging from 1.3 to 20, compared to $\kappa_{all}$, indicating that $V_{coating}/V_{BC}$ has a
substantial influence on $\kappa_{BC}$ (Fig. 2d). This can be evidenced by the maximum value of $\kappa_{BC}$ correlates with that of
$V_{coating}/V_{BC}$ (Fig. 2c and Fig. 2d), and the divergence of the maximum values of $\kappa_{all}$ and $\kappa_{BC}$ (Fig. 2d). The coating
of BCc is formed through by the photo-chemistry of VOCs precursors during the atmosphere aging process,
which is determined by VOCs precursors and also aging time. Comparing to the clean clusters C3 and C4, the air
mass movement of C1 was much slower, combined with passing the highest pollution level region, the slowly
transported air mass from the polluted southern region was characterized with a longer transport time than other
air mass clusters (Fig. 1b). As expected, the observation of PTR-ToF-MS shows higher VOCs values and also the
longer $t_{age}$ during C1 polluted condition (Fig. S2). Figure 2i show that the $t_{age}$ during C1 polluted period (13h) was
about one times larger than that during C3/C4 clean period (9h), this reflected the air during polluted period was
more aged.
Figure 3 shows diurnal variation of $PM_1$, BC mass concentration, OA fraction, $V_{coating}/V_{BC}$, $\kappa_{all}$ and $\kappa_{BC}$ for the
different air masses respectively. More pronounced diurnal patterns were found during the C3/C4 clean period, as
the fast-moving clean air masses dilute the pollutant and were mainly influenced by local emissions, however,
regional transport and secondary aerosol formation influence the diurnal pattern during C1 and C2 period (Fig.
2a). For chemical inert BC particles, clear BC mass peaking at morning and afternoon rush-hour (marked as grey
bar) (Fig. 3b) was observed which is typical time period of heavy traffic with nascent BC emitted, but C1 period
showed a continuous transport and increased BC mass loading throughout the midday, even when developed
planetary boundary layer (PBL) dilute the pollutants.
Fresher BCc showed lowest $V_{coating}/V_{BC}$ during the morning rush-hours, and within a few hours the $V_{coating}/V_{BC}$
quickly increase and peak at noon when solar radiation was strongest for all-time period (Fig. 3d). The increase
during midday was in consistent with the OA fraction (Fig. 3c), which suggested that the acquire coating of BC
particles might be secondary OA through the photochemical oxidation, inconsistent with recently studies (Cui et
al., 2022; Wu et al., 2019). As enhanced OA fraction at noon due to photochemical SOA formation (Fig. 3c),
result a lower value of $\kappa_{all}$ during noontime (Fig. 3e). $\kappa_{BC}$ is more influenced by the $V_{coating}/V_{BC}$, the $\kappa_{BC}$ was found
increase after morning rush-hours due to the increase of coating on BC.

### 3.3. The CCN Activation Properties of All-particles and BCc

### 3.3.1 Determining the D50 of BCc

The measurement of the activation properties of all-particles and BCc into CCN was discussed in Sec. 2.3.2.
Figure 4 illustrates a representative example of size-resolved activation of all-particle and BCc at SS = 0.1% and
0.3% respectively. Activation commences for all-particles when [$CCN$] is greater than 0 (depicted by blue lines),
while BCc (indicated by black lines) begins to activate when the [$CN_{unact}$] (represent by green lines) falls below
that of [$BCc$] (Fig. 4). [$BC_{act}$] (denoted by the grey shadow) can be derived using Equation 4, and the size-
resolved $F_{act,BC}$ can be calculated in accordance with Equation 5. It can be seen that with the increase of particle
size, BCc started to get activated at 0.19μm (0.13μm) at SS=0.1% (0.3%), which is higher than all-particle by 44%
(46%), the higher D50 of BCc also indicates that the lower $\kappa_{BC}$ than $\kappa_{all}$.
We would like to emphasize that the principal uncertainty in the method employed stems from the assumption
regarding the lower $\kappa$ for BC compared to other particles. The assumption was consistent with concurrent
theoretical understanding, pure BC was hydrophobic ($\kappa_{BC}$=0), though the $\kappa_{BC}$ increase by acquire hygroscopic
coating through atmospheric aging (Grimonprez et al., 2018; Weingartner et al., 1997), the $\kappa_{BC}$ remains lower
than other components under the same atmospheric oxidation process based on ZSR volume mixing rule. Most
importantly, even though some internally mixing BCc activate into CCN previous to other chemical components,
the results here was underestimated the [$BC_{act}$] and $F_{act,BC}$, so the CCN activity of BCc derived through this
method could represent the low bound of the atmospheric BCc.



### 3.3.2 D50 of BCc and All-particles

Figure 5 illustrate the temporal evolution of size distribution of BCc, CN and CCN, respectively. The [*CCN*] increase during polluted condition due to the increase BCc and all-particles concentration, and the [*CCN*] is mainly contributed by all-particles (Fig. 5c). The derived D50 for BCc and all-particles under water SS at 0.1%, 0.2%, and 0.3% is shown as lines in Fig. 5a-c for the temporal evolution, and mean value is shown as vertical dash lines in Fig. 5d. The mean D50 of BCc particles (0.17μm, 0.13μm, and 0.12μm under SS= 0.1%, 0.2%, and 0.3%) was approximate 1.4 times larger than that of all-particles (0.11μm, 0.08μm, and 0.07μm under SS= 0.1%, 0.2%, and 0.3%) (Fig. 5), which was expected from $\kappa$-Köhler theory (Petters and Kreidenweis, 2007) that lower/higher D50 means the higher/lower $\kappa$. The higher D50 of BCc also reflected lower $\kappa$ and BCc should be activated in the last order.

Large variation of D50 of BCc and all-particles was also found for different air massed (Fig. 5), the D50 of all-particles (0.13μm ± 0.01μm) and BCc (0.20μm ± 0.01μm) during C1 polluted period was lower that (0.15μm ± 0.02μm) and (0.23μm ± 0.02μm) during C3/C4 clean period. This is also inconsistent with the variation of $\kappa_{all}$ and $\kappa_{BC}$ (discussed in 3.2), suggested the more inorganic fraction increase $\kappa_{all}$ and the thick coating increase the $\kappa_{BC}$ during the polluted condition.

Figure 6a and 6b illustrate the diurnal variation of D50 of all-particles and BCc under SS=0.10%, the other two SSs at 0.20% and 0.30% are shown in Fig. S3 and Fig. S4. The diurnal variation was more pronounce under clean period (i.e. C3+C4), when local emission dominate the pollution sources. Higher D50 of all-particles and BCc was found during the rush-hours when more primary hydrophobic OA and fresh external mixed BCc emitted. The D50 of all-particles and BCc will decrease after photochemistry oxidation secondary OA formation and coating on BCc which was also in consistent with that of $\kappa_{all}$ and $\kappa_{BC}$ (Fig. 3).

### 3.3.3 Droplet Activation Fraction of BCc and All-particles

Figure 5d shows the averaged size distribution of BCc, CN and CCN under three SSs for different periods classified by air mass clusters (C1-C4). Notably BCc had consistently larger peak size than all-particles for all air masses. For example, BCc peak from 185 to 210nm, which was about 2-4 times larger than that of all-particles, which peaked from 58 to 115nm. The more polluted air masses (C1 and C2) had larger peak size of BCc and CN than cleaner air masses (C3 and C4), because of the condensation process during the polluted episode when gases heavily partitioned to aerosol phase and enlarged particle size.

Though the BC core size is smaller, after attaching on larger particles the overall coated size is even significantly larger than BCf. This means in the typical sub-urban environment, BC had been significantly associated with non-BC substance through condensation and coagulation process (Yu et al., 2022), and the bare BC without containing non-BC substance had taken a less proportion and most BCc had been enlarged during the aging process. This is consistent with previous studies that BC was associated with larger aerodynamic size than BCf (Pan et al., 2019; Wang et al., 2021). Previous studies also observed rapid aging of BC in a few hours in urban environment (Peng et al., 2016).

Figure 6d and 6e show the diurnal variations $F_{act}$ (in number and mass) for all-particles and BCc under SS=0.1%. The other two SSs at 0.2% and 0.3% are shown in Figure S2 and Figure S3. With SS increase from 0.1% to 0.2%, the number $F_{act,BC}$ and $F_{act,all}$ increased from 42%±15% to 69%±15% and from 30%±15% to 51%±19%. Figure 6f show the statistical $F_{act}$ results compare for different air masses. the number $F_{act,BC}$ increased from 30% to 53% under SS=0.1% (Fig. 6f) with the pollution increase from C3/C4 to C1. The mass $F_{act,BC}$ show similar variation characteristic as the number $F_{act,BC}$, which increase from 58 % to 76 % under SS=0.1% from C3/C4 to C1 period (Fig. 6f), and from 81% to 89% and from 87% to 91%, under SS at 0.2% and 0.3%, respectively. BCc acquires thicker hygroscopic coating through longer atmospheric aging time during the long range regional transport C1 polluted period, the hygroscopic coating not only increase $\kappa_{BC}$ and lead to a lower D50, also increasing the BCc



size. The decrease of D50 and increase of particle size both led to an increase of $F_{act}$. The $F_{act,BC}$ mainly depended
on the $V_{coating}/V_{BC}$, as the D50 and size of BCc was dependent on $V_{coating}/V_{BC}$. The diurnal variation also show
lowest $F_{act,BC}$ during rush hours due to lower $V_{coating}/V_{BC}$, and increased in the daytime with increased $V_{coating}/V_{BC}$
(Fig. 6e).
The number $F_{act,BC}$ was always higher than $F_{act,all}$ under all SSs (Fig. 6, Fig. S2 and Fig. S3), due to the size of
BCc is larger than all-particles. The mass $F_{act,BC}$ was about 1.61 higher than in number, compared with about 2.88
difference of mass and number $F_{act}$ for all-particles. This is also caused by the larger size distribution of BCc than
all-particle. The large portion of all-particles in small size drives the number $F_{act}$ to be lower than BCc. Although
D50 of BC is higher, the larger size distribution of BCc than all-particles means BCc can have a higher droplet
removing efficiency than BCf.
The large $F_{act,BC}$ than BCf particles, and thickly coated BCc have preference in CCN activation was also observed
through cloud residues observation at a mountain site in south China (Fu et al., 2022). The measured $F_{act}$ was
generally consistent with experiment conducted in a South China mega city using the same method (Hu et al.,
2021a), but are higher than cloud residues observation in China (Zhang et al., 2017), which reported an average
value 33% of the mass $F_{act,BC}$ and number $F_{act}$ of BCc (all-particle) ranged in 5%-45% (7%-60%). We attribute
the difference to unknow ambient SS for the cloud, which have great influence on the $F_{act,BC}$. According to
previous cloud residues observation at Jungfraujoch, the number $F_{act,BC}$ increase from 50% to 80% when SS
increase from 0.21% to 0.50% (Motos et al., 2019a; Hammer et al., 2014). The ambient SS show large variation
can be range from 0.01% to 2.00% (Hammer et al., 2014), however, it is hardly to direct observe the SS from the
field observation. Through this method the number and mass $F_{act,BC}$ activated into droplets can be quantitatively
investigate under certain ambient SS condition, thus the results here provide a reference to investigate the lifetime
of BCc.
### 3.4. Time Scale for the Evolution of CCN Activity of BCc
The temporal evolution of parameters related to CCN activity including $\kappa$, D50 and $F_{act}$ against photochemical
age ($t_{age}$) for BCc and all-particles are shown in Fig. 7. It shows that the aging time ($t_{ag}$) can well explain the
variation of all parameters for BCc and all-particles with a linear function ($R^2>0.4$).
The volume ratio of coating over rBC ($V_{coating}/V_{BC}$) showed an increase rate of 0.14 h$^{-1}$ (Fig. S4), and can be fitted
as:
$$V_{coating}/V_{BC} = 0.55 + 0.14 \cdot t_{age} \qquad (7)$$
This results in a linear increase rate for the hygroscopicity parameter, $\kappa_{BC}$ showed a linear rate of 0.09 h$^{-1}$ ((Fig.
7a, 7b, and 7c)) at all SSs, can be fitted as:
$$\kappa_{BC} = 0.09 + 0.01 \cdot t_{age} \qquad (8)$$
In addition, $\kappa_{BC}$ also follow a linear function with $t_{age}$ as:
$$\kappa_{all} = 0.16 + 0.014 \cdot t_{age} \qquad (9)$$
This means the $\kappa$ for the large proportion of non-BC is enhanced after aging due to enhanced formation of
inorganic salt and secondary organic aerosols (Wu et al., 2022). The results show it will take about 24 and 10
hours to reach the global mean $\kappa$ of 0.27-0.30 (Pringle et al., 2010) for BCc and all particles, respectively.
The aging scale of a few hours observed here is generally consistent with previously measured BC aging
influenced by anthropogenic pollution, such as an aircraft measurement (Moteki et al., 2012) which observed the
conversion time scale of BC was a few hours under polluted environment. The results were also comparable with
previous chamber studies which investigated the $\kappa$ evolution of BC (Tritscher et al., 2011; Lambe et al., 2015;





Peng et al., 2017), the increase rate of $\kappa_{BC}$ (0.01h$^{-1}$) in this study was similar, but the initial $\kappa_{BC}$ here (0.08-0.26)
was higher than the fresh combustion soot as used in their results (0-0.12). These consistently showed a few
hours tended to be a typical aging time scale to convert appreciable amount of initially hydrophobic BC to be
hydrophilic, hereby CCN active. The exact time scale may according to pollution level, such as a higher
concentration of gas precursors and favoured meteorological condition may promote the growth rate of
hygroscopic substances on BC.
Correspondingly, the D50 of BCc particles and all-particles decreased with $t_{age}$, with a rate of -3 nm h$^{-1}$ and -3nm
h$^{-1}$(Fig. 7d), -1 nm h$^{-1}$ and -2nm h$^{-1}$ (Fig. 7e), and -0.5 nm h$^{-1}$ and -2 nm h$^{-1}$(Fig. 7f) under SS = 0.10%, 0.20%,
and 0.30%, respectively. The results are summarized in Table 1.
The number activation fraction $F_{act,all}$ and $F_{act,BC}$ also show linear corelation with $t_{age}$. A notable higher CCN
activation fraction was observed for BCc than all particles at the same $t_{age}$, this was attributed to the larger coated
size of BCc than other non-BC particles (Fig. 5). This difference between BCc and other particles was reduced at
longer $t_{age}$ because the sufficient aging time had allowed all particles to grow to similar large sizes.
After 8 hours of aging, $F_{act,BC}$ increased from 35% to 63% as SS rose from 0.1% to 0.2%, and then increase from
63% to 77% as SS increase from 0.2% to 0.3%. This illustrates a notable increase in $F_{act,BC}$ from SS=0.1% to
0.20%, but a less significantly changed $F_{act,BC}$ from SS=0.2% to 0.3%. Furthermore, the rate increase of $F_{act,BC}$
with $t_{age}$ was 2% h$^{-1}$ at SS=0.10% (Fig.7g), which was twice as large as at SS=0.2% (1% h$^{-1}$) (Fig. 7h) and 0.3%
(1% h$^{-1}$) (Fig. 7i). This indicated that the activation of BCc was more sensitive at lower SS (<0.2%), but may
reach a plateau when SS>0.2%. This is because the activation diameter at this typical SS range was close to the
mode size of BCc number distribution (Fig. 5d), and a small increase of SS will cause a considerable change of
fraction which is larger than the activation diameter. The SS range from 0.1% to 0.2% is for the typical formation
of fog and stratiform thin clouds (Hammer et al., 2014).
**4. Conclusion**
In this study, we conduce directly measurements of the droplet activation of BC-containing particles under water
supersaturation conditions in a sub-urban environment. Both the hygroscopicity and CCN activation diameters
reveal that the hygroscopicity of all-particles and BCc concurrently increase with polluted levels, primarily
governed by the amounts of coating. As pollution intensifies, the volume ratio of coating over rBC increases from
1.8 to 4.7, causing the hygroscopicity of BCc to increases from 0.11 to 0.23. The increase of coating not only
increase BCc's hygroscopicity but also shifts these particles to larger sizes, significantly enhance their CCN
activity. The number activated fraction of BCc was 42% under SS at 0.1%, higher than the 30% activation of all
particles. Despite the lower hygroscopicity, the elevated activation fraction of BCc can be attributed to their
larger size compared to all particles. It was also found that the number activated fraction of BCc was more
sensitive in the SS range from 0.1% to 0.2%, a typical range for fog and stratiform thin clouds formation. This
means a small increase of SS even for the environment with relatively low moisture and weak updraft the BC can
be activated. Considering these clouds have a lower precipitation rate, BC in these clouds may have more
persistent indirect radiative impacts.
Furthermore, we successfully parameterized the evolution of hygroscopicity and CCN activation characteristics
of BCc and all particles in relation to equivalent photochemical age. The hygroscopicity of BCc was found
positive linearly correlated with aging time with a rate of 0.01 h$^{-1}$, and the activation diameter negative correlated
with a rate of 1-3 nm h$^{-1}$. Within 10 hours, 65% of BCc activated under SS=0.2% in this typical sub-urban
environment, this is in line with a wide range of recent ambient studies that BC can be half activated under



SS=0.1-0.2%. This means these BCc may be efficiently incorporated into clouds by serving as CCN. This aging
process by acquiring more hygroscopic substances of BC may be particularly important under polluted
environment, where the condensation and coagulation processes intensively occurred. The results here thus
provide direct evidence of efficient droplet activation of BCc, and the aging scale associated with their CCN
activation can be used as a refence to validate the regional models regarding aerosol-cloud interactions of BC.
**Data availability.** All data in this paper are available from the authors upon request (tianping@bj.cma.gov.cn).
**Competing interests.** At least one of the (co-)authors is a member of the editorial board of Atmospheric
Chemistry and Physics.
**Author contribution.** D. D., and M. H. led and designed the study; P. T. and D. L. designed the study, set up the
experiment, analysed the data, and wrote the paper. K. H. and Y. W. conducted the field observation and
contributed the data analysis. H. H. and J. S performed the calibration of the instruments. C. Y. and D. H
contributed to scientific discussion.
**Acknowledgment.** This research was supported by the National key Research and Development Program of
China (2023YFC3007600), and National Natural Science Foundation of China (42175116, 41975180, 42275087).



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



Table 1. A summary of linear fitting results  for activation diameter (D50) and activation fraction ($F_{act}$) of all
particles and BCc-containing particles.

| Initial, Slope | Activation diameter (D50) | | Activation fraction ($F_{act}$) | |
|---|---|---|---|---|
| | All particles | BC-containing particles | All particles | BC-containing particles |
| SS=0.1% | 170nm, -3nm h$^{-1}$ | 240nm, -3nm h$^{-1}$ | 4%, 2% h$^{-1}$ | 19%, 2% h$^{-1}$ |
| SS=0.2% | 120nm, -2nm h$^{-1}$ | 170nm, -1nm h$^{-1}$ | 18%, 0.3%h$^{-1}$ | 55%, 0.1% h$^{-1}$ |
| SS=0.3% | 100nm, -2nm h$^{-1}$ | 150nm, -0.5nm h$^{-1}$ | 26%, 0.3% h$^{1}$ | 69%, 0.1% h$^{-1}$ |




## Figures and captions

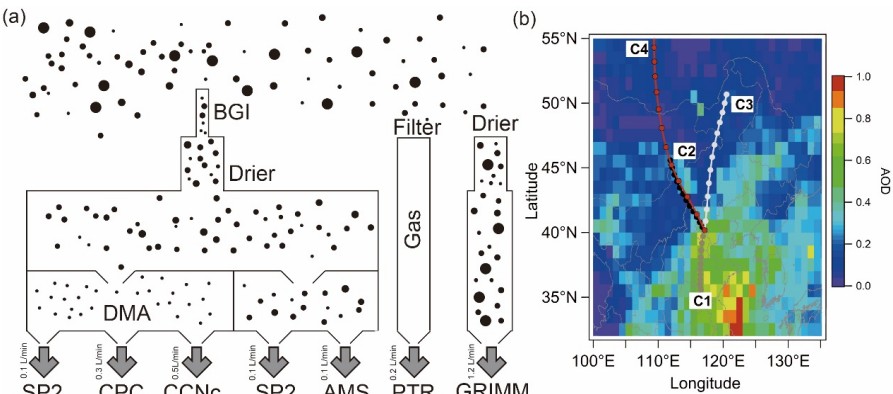

**Figure 1.** Schematic of the experiment. a) Experimental setup showing aerosols are sampled through a PM$_{2.5}$ impactor following a drier. Mono-dispersed aerosol was measured downstream of a differential mobility analyser (DMA), including SP2, CPC and CCNc, and additional line measure the poly-dispersed aerosols with SP2 and HR-ToF-AMS. All sizes of dried aerosol are measured by a GRIMM, and the VOC concentration was measured by a PTR-TOF-AMS. b) Mean aerosol optical depth (AOD) during the experimental month, with three lines showing the clustered 36h backward trajectories initialized from the experimental site with each markers denoting 6h transport time.



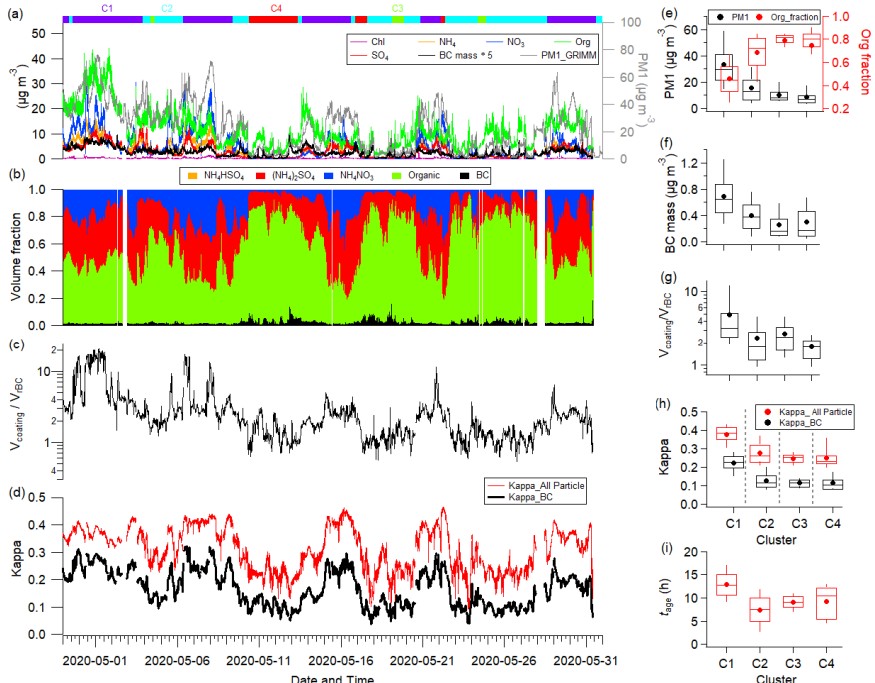

**Figure 2.** Time series of a) mass concentrations for chloride (Chl), ammonium (NH$_4$), nitrate (NO$_3^-$), organic (Org), sulfate (SO$_4^{2-}$), BC mass and PM$_1$ measured by HR-TOF-AMS, SP2 and GRIMM. b) Volume fraction of particle components for NH$_4$HSO$_4$, (NH$_4$)$_2$SO$_4$, NH$_4$NO$_3$, Organic and BC, c) V$_{coating}$/V$_{BC}$, d) $\kappa$ for all particle and BC respectively. The flag on the top of the graph represents different clusters by different colours. Mass concentrations of e) Organic fraction and PM1 by summing HR-ToF-AMS components and BC mass from SP2, f) BC mass, g) V$_{coating}$/V$_{BC}$, h) $\kappa$, i) $t_{age}$ for all particle (red) and BC (black) for four clusters, with solid circle indicating the average; in each box, the solid line and top and bottom boundaries represent the median, 75$^{th}$ and 25$^{th}$ percentiles, respectively.



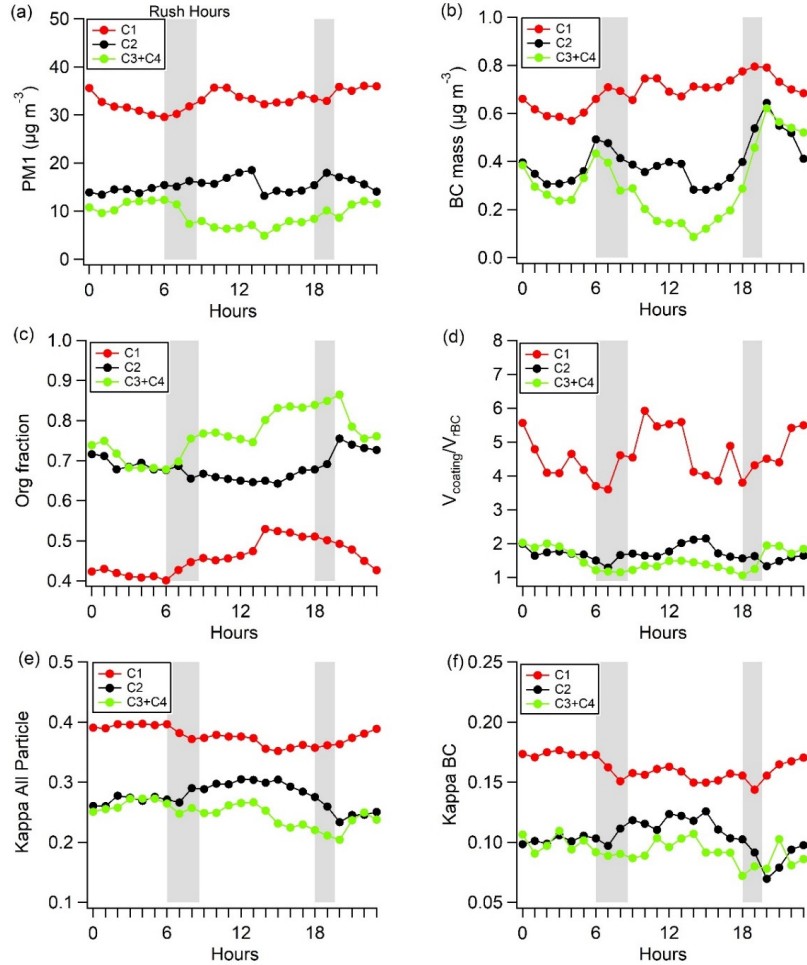

662

**Figure 3.** Diurnal variations of a) PM$_1$, b) BC mass, c) Organic fraction, d) V$_{coating}$/V$_{BC}$, e) $\kappa$ for all particles and f) $\kappa$ for BC for four clusters, respectively. Grey bars indicate the traffic rush hours, and the red, black, and green line represent the C1, C2 and C3/C4 period, respectively.

666



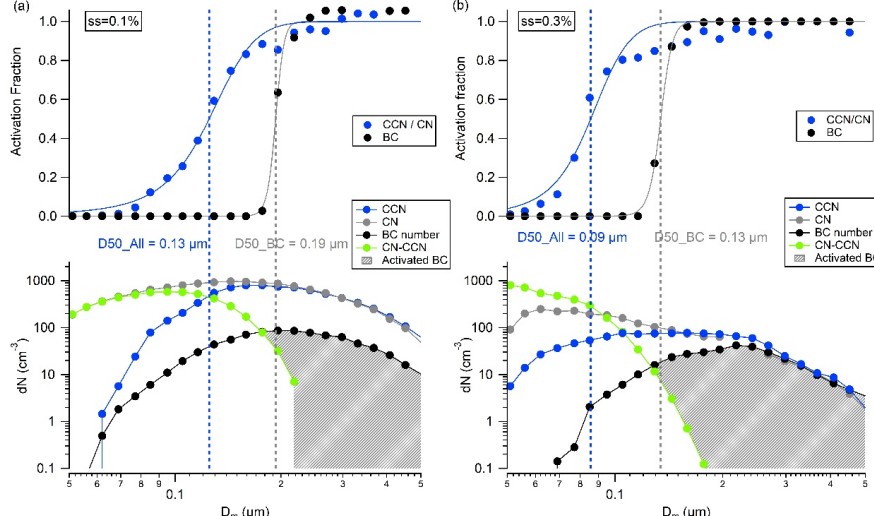

**Figure 4**. A typical example of size-resolved number concentrations of CN, CCN and BCc under supersaturation of a) 0.1% and b) 0.3%. The green lines show the un-activated number concentration and the grey shadows indicate the activated BCc number concentrations at each size. The blue and black dots in the upper panel show the activation fraction of all-particle and BCc. The blue and grey lines indicate the sigmoid fitting on the size-resolved activation fraction of all-particle and BC respectively, with the vertical dash lines indicating the respective activation diameters.



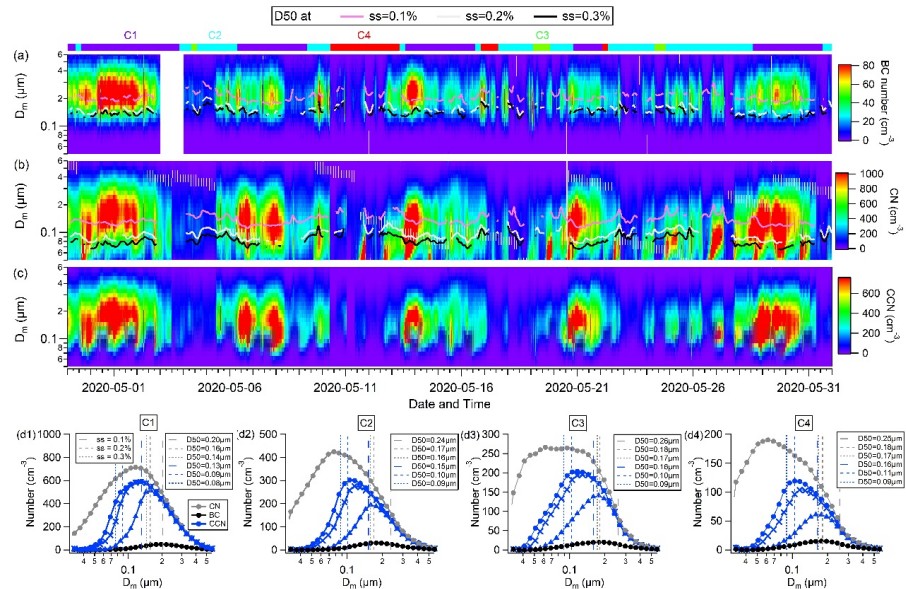

**Figure 5.** Temporal evolution of numbers of a) BCc, b) CN, c) CCN and activation diameter (D50) of BCc and CN particles under SS=0.1%, 0.2% and 0.3%. d) Mobility particles size distribution of CN, BCc and CCN numbers under SS=0.1%, 0.2% and 0.3% for different clusters. The blue and grey dash lines on the images denote the D50 for all-particles and BCc under different SS, respectively. Bottom panels show the mean particles size distributions of CN, CCN (under the three SSs) and BCc, for the classified air masses.

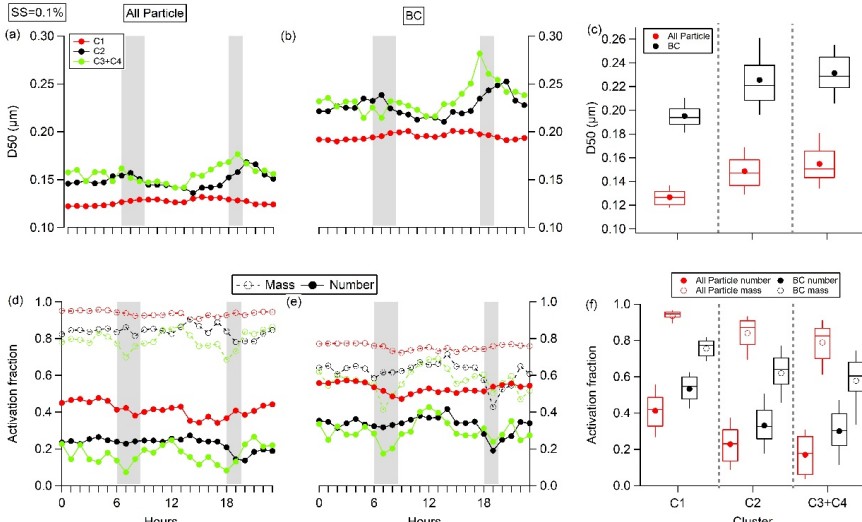

**Figure 6.** Diurnal variations of D50 of a) all-particles, and b) BCc particles under SS=0.1% for different air mass clusters, respectively, c) the statistical of D50 for different air mass cluster, with solid circle indicating the average; in each box, the solid line and top and bottom boundaries represent the median, 75[th] and 25[th], respectively. c) diurnal variation of the activation fraction of all-particles, and e) BCc particles, with the solid and dash line represent the number and mass fraction, the mass (number) activation fraction was calculated by summing the mass (number) of particles with diameter larger than D50, and divided by the total mass (number). f) statistical of the activation fraction.



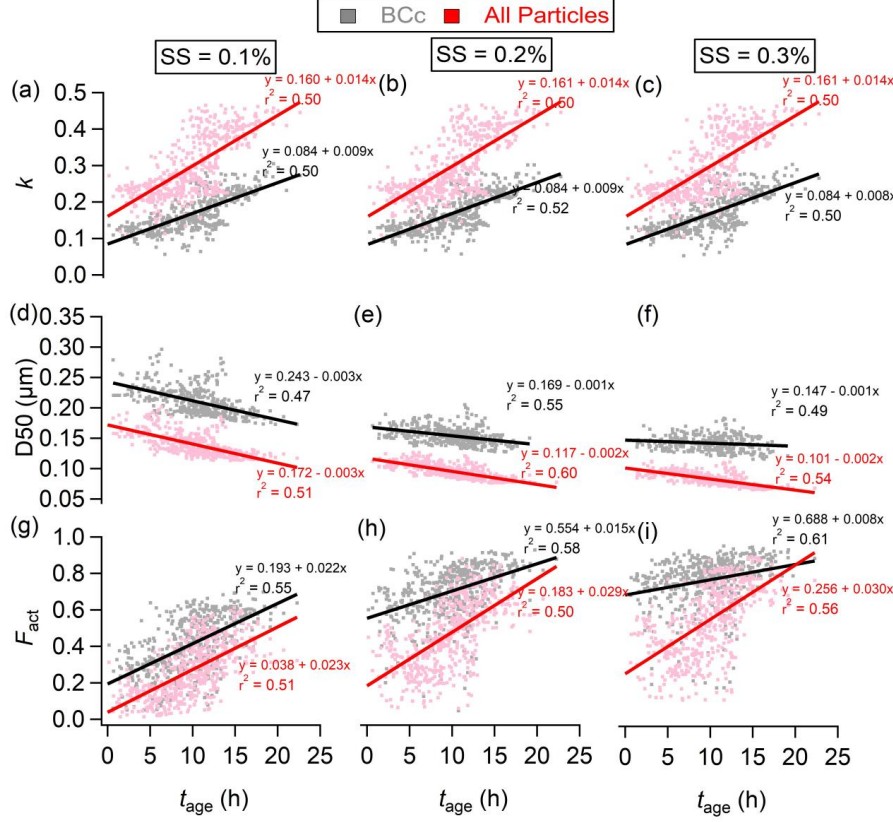

**Figure 7.** The evolution of hygroscopicity parameter ($\kappa$) (a-c), activation diameter (D50) (d-f), and number activation fraction ($F_{act}$) (g-i) with photochemical age ($t_{age}$) under SS at 0.1%, 0.2%, c) 0.3%. The black and red denote the BCc particles and all-particles, respectively. The lines show the least-square lineal regression.