# Peer review of "Efficient droplet activation of ambient black carbon particles in sub- # 2 urban environment"

_EGUsphere, 2023_

## Referee Comment (RC1)

The author used the tandem system of DMA-CCN coupled with SP2 to directly monitor the droplet activation diameter (D50) and activation fraction of BCc in ambient environment. Further, the author also quantifies activation properties of BC on a time scale, and found BCc from anthropogenic sources can readily serve as CCN at a relatively low SS. I believe that the topic is interesting and it could be useful to the scientific community. However, some modifications are needed before it can be accepted.

Major comments:

1. The author calculates the photochemical age by the toluene/benzene ratio. However, even during the day, toluene can be affected by air mass transport, local emissions, or solvent volatilization, which introduces considerable uncertainty to the calculation. Can the author give more information support your result?
2. Lines 233-238, As far as I know, the chemical composition on BC was different with that of all particles, which may lead to the deviation of the $\kappa$ of BC calculated based on this method, can the author further elaborate on this?

Specific comment:
Line 196: xxx is the ration of measured ……., may be ratio? Rather than ration.

---

## Author Comment (AC1)

We thank the reviewers for their comprehensive and insightful comments on our manuscript, which have greatly improved the manuscript. We have addressed all the comments reviewers raised in the revision. According to reviewers' comments, the main revisions we made include:

1. We have incorporated the suggestions provided by the reviewers to improve the introduction and discussion of the measurement techniques used in our study. We have elaborated on the technical details and methodology.

2. We have carefully reviewed the manuscript for editorial and grammatical errors and corrected these issues to improve the readability of our work.

**Reviewer 1:**

General comments:

  The author used the tandem system of DMA-CCN coupled with SP2 to directly monitor the droplet activation diameter (D50) and activation fraction of BCc in ambient environment. Further, the author also quantifies activation properties of BC on a time scale, and found BCc from anthropogenic sources can readily serve as CCN at a relatively low SS. I believe that the topic is interesting and it could be useful to the scientific community. However, some modifications are needed before it can be accepted.

We thank the reviewer for the positive comments and suggestions. We have carefully considered your feedback and have made the following revisions to our manuscript.

**Major comments:**

1. The author calculates the photochemical age by the toluene/benzene ratio. However, even during the day, toluene can be affected by air mass transport, local emissions, or solvent volatilization, which introduces considerable uncertainty to the calculation. Can the author give more information support your result?

We acknowledge the reviewer's concern regarding the potential uncertainties in calculating the equivalent photochemical age ($t_{age}$) due to the mixing of air masses. However, this methodology, despite its limitations, has been validated as a reasonable metric to evaluate the photochemical age in the region with slimmer and stable source emissions. The influence of adding a significant proportion of local fresh air does affect the measured toluene/benzene ratio, yet it concurrently indicates that such air masses are relatively less aged compared to those that have undergone regional transport. The application and the uncertainty of using $t_{age}$ has been given in our prior publication (Wu et al., 2022), which we have directly applied in this study. We have added the related discussions in the revision.

Line 188-206:

"The mass spectra of VOCs were measured using the PTR-TOF-AMS, calibrated with the CH3COCH4+ (m/z 59.0490) and H3O+ (m/z 21.0226), following the procedures

described by Bruns et al. (2016). The equivalent photochemical age ($t_{age}$), a critical factor for assessing the duration of atmospheric photochemical reaction, is derived from comparing the ratios of two VOCs with different OH radical reaction rates, acknowledging that species more reactive with OH diminish more rapidly as photochemical aging increases (Yuan et al., 2012). Despite uncertainties arising from the mixing of fresh emitted and aged air masses, this method provides valuable insights into atmospheric photochemical processing (Parrish et al., 2007). In this study, toluene and benzene were chosen for their strong correlation and differing reaction rates with OH, and expressed as:

$$t_{age} = \frac{1}{[OH](k_{toluene} - k_{benzene})} \times \left[ \ln\left(\frac{[toluene]}{[benzene]}\right)_0 - \ln\left(\frac{[toluene]}{[benzene]}\right) \right] \tag{6}$$

where [OH] is the ambient average OH concentration ($2 \times 10^6$ molecule cm$^{-3}$) obtained from the reanalysis product (available from the European Center for Medium Range Weather Forecasts: Atmospheric Composition Reanalysis 4), consistent with previous observation in Beijing (Liu et al., 2018). $k_{toluene}$ and $k_{benzene}$ is the reaction rate with OH of toluene ($5.63 \times 10^{-12}$ cm$^3$ molecule$^{-1}$ s$^{-1}$) and benzene ($1.22 \times 10^{-12}$ cm$^3$ molecule$^{-1}$ s$^{-1}$), respectively. $(\frac{[toluene]}{[benzene]})_0$ is the ratio of freshly emitted toluene and benzene, and the value was determined to be $2.27 \pm 0.27$ according to the maximum toluene/benzene ratio during the experiment. $\frac{[toluene]}{[benzene]}$ is the ratio of measured toluene and benzene in the atmosphere. The calculation of $t_{age}$ was performed only during daytime ensure it reflects the photochemical process. The dataset of $t_{age}$ in this study has been successfully tested to determine the photochemical reaction or production rates of secondary OA, with further details on the calculation of [OH] and $\frac{[toluene]}{[benzene]})_0$ available in Wu et al., (2022). "

2. Lines 233-238, As far as I know, the chemical composition on BC was different with that of all particles, which may lead to the deviation of the $\kappa$ of BC calculated based on this method, can the author further elaborate on this?
Here we consider the compositions of substances either internally or externally mixed with BC are similar, which are all from the AMS measurement. This approximation has been used in many previous studies (Liu et al., 2013; Ohata et al., 2016) for the best estimate for the coatings on BC. Though some studies found the coatings on BC contained a higher OA fraction than BC-free particles (Wang et al., 2020). Considering the OA has a lower $\kappa$ than other inorganic salt, this may lead to an overestimation of $\kappa$ of BCc. These discussions are now added in the revision.

Line 247-250:
"Many studies assumed the same components between BC coatings and other substances externally mixed with BC (Liu et al., 2013; Ohata et al., 2016), as consistent with the assumption in this study. However, the coatings on BC may contain a higher OA fraction than other BC-free particles (Wang et al., 2020), which means the current

calculation may give a higher estimation of $\kappa_{\text{BCc}}$."

**Specific comment:**

Line 196: xxx is the ration of measured ......., may be ratio? Rather than ration.

Corrected.

Line 202-203:

"$\dfrac{[toluene]}{[benzene]}$ is the ratio of measured toluene and benzene in the atmosphere."

Reference:

Liu, D., Allan, J., Whitehead, J., Young, D., Flynn, M., Coe, H., McFiggans, G., Fleming, Z. L., and Bandy, B.: Ambient black carbon particle hygroscopic properties controlled by mixing state and composition, Atmos. Chem. Phys., 13, 2015-2029, 10.5194/acp-13-2015-2013, 2013.

Ohata, S., Schwarz, J. P., Moteki, N., Koike, M., Takami, A., Kondo, Y.: Hygroscopicity of materials internally mixed with black carbon measured in Tokyo, J. Geophys. Res. Atmos., 121, 362-381, 2016.

Cui, S., Huang, D. D., Wu, Y., Wang, J., Shen, F., Xian, J., Zhang, Y., Wang, H., Huang, C., Liao, H., and Ge, X.: Chemical properties, sources and size-resolved hygroscopicity of submicron black-carbon-containing aerosols in urban Shanghai, Atmos. Chem. Phys., 22, 8073-8096, 10.5194/acp-22-8073-2022, 2022.

Parrish, D. D., Stohl, A., Forster, C., Atlas, E. L., Blake, D. R., Goldan, P. D., Kuster, W. C., Gouw, J. A. de.: Effects of mixing on evolution of hydrocarbon ratios in the troposphere, J. Geophys. Res. Atmos., 112, D10S34, 2007.

Wu, Y., Liu, D., Tian, P., Sheng, J., Liu, Q., Li, R., Hu, K., Jiang, X., Li, S., Bi, K., Zhao, D., Huang, M., Ding, D., and Wang, J.: Tracing the Formation of Secondary Aerosols Influenced by Solar Radiation and Relative Humidity in Suburban Environment, J. Geophys. Res. Atmos., 127, 10.1029/2022jd036913, 2022.

Wang, J., Ye, J., Liu, D., Wu, Y., Zhao, J., Xu, W., Xie, C., Shen, F., Zhang, J., Ohno, P. E., Qin, Y., Zhao, X., Martin, S. T., Lee, A. K. Y., Fu, P., Jacob, D. J., Zhang, Q., Sun, Y., Chen, M., and Ge, X.: Characterization of submicron organic particles in Beijing during summertime: comparison between SP-AMS and HR-AMS, Atmos. Chem. Phys., 20, 14091–14102, https://doi.org/10.5194/acp-20-14091-2020, 2020.

---

## Author Comment (AC2)

We thank the reviewers for their comprehensive and insightful comments on our manuscript, which have greatly improved the manuscript. We have addressed all the comments reviewers raised in the revision. According to reviewers' comments, the main revisions we made include:

1. We have incorporated the suggestions provided by the reviewers to improve the introduction and discussion of the measurement techniques used in our study. We have elaborated on the technical details and methodology.

2. We have carefully reviewed the manuscript for editorial and grammatical errors and corrected these issues to improve the readability of our work.

**Reviewer 2:**

Review of "Efficient droplet activation of ambient black carbon particles in sub-urban environment" by Tian et al.
This study observationally investigated the hygroscopicity and CCN activity of BC-containing particles in a suburban environment, with particular attention to the source-dependent variability and the evolution with photochemical-age. To my knowledge, quantitative observations of the relationship between the activated number-fraction of BC-containing particles and photochemical age have been rare or have been conducted for the first time. Particularly, Figure 7 results should be useful to constrain/validate the aerosol microphysical models as well as the BC-aging schemes adopted in the aerosol-climate models. I can recommend this manuscript for publication after incorporating the following request for minor corrections.

We are thankful for the positive comments of reviewer.

**General comments:**

This manuscript contains many editorial or grammatical errors. The authors should check the manuscript more seriously before submission. In addition, the authors should thoroughly check again the appropriateness of the cited references in each sentence.

We have carefully reviewed the manuscript for editorial and grammatical errors and believe we have corrected these issues to improve the readability of our work. The appropriateness of the cited references has been also rechecked to ensure accuracy and relevance to our claims and findings.

**Minor/Individual comments:**

L25: Specify the location/observation site.

The location was added.

Line 22-25:

"Here, we directly measured the droplet activation diameter (D50) and activation fraction of BCc ($F_{act,BC}$) in suburban Beijing using coupled measurements of sizeresolved number concentrations of CCN at configured water supersaturation (SS) and BC-containing particles (BCc)."

L42-43: Check the grammatical correctness of this sentence.

This sentence was rewritten.

Line 42-44:

"Wet scavenging, identified as the primary removal mechanism for atmospheric BC (Jacobson, 2010), involves BC particles acting as cloud condensation nucleation (CCN) that are activated into the cloud droplets and removed through precipitation."

L52: For clarity, I suggest: "characterizing" -> "predicting".

Corrected.

Line 52-53:

"has been successful in predicting the hygroscopicity of internally mixed BC, assuming a spherical core-shell structure."

L69: Remove "in east Asian polluted area".

Corrected.

Line 67-69:

"More than 50% of BC mass can be removed through the wet removal process during transport from the surface to the planetary boundary layer (PBL), as evidenced by studies in East Asian polluted areas"

L74: "sub saturation" -> "subsaturation"

Corrected.

Line 74:

"and the hygroscopic tandem differential mobility analyser (HTDMA) under subsaturation condition."

L78: Reword "how many parts of BC". It sounds awkward to me.

This sentence was rewritten.

Line 76-77:

"but it cannot distinguish how much of the BC contributes to activation, given that BC accounts for about 10% of the ambient aerosol."

L85: Reword "air mass clusters" -> "air mass types".

This sentence was rewritten.

Line 83-84:

"The variation in the hygroscopicity of BC during different pollution levels was investigated through the classification of four distinct air masses types."

L85: "based on the" -> "based on the trace-gas measurements by the"

This sentence was rewritten.

Line 84-85:

"The equivalent photochemical age ($t_{age}$), calculated using trace-gas measurements by the Proton-Transfer-Reactor Time-of-Flight Mass Spectrometer,"

L90: "for more than 1 month" -> "for XX days"

Corrected.

Line 89:

"The field campaign was conducted for 33 days from 29th Apr. to 2nd Jun. 2020"

L92: "65Km" -> "65 km"

Corrected.

Line 90-91:

"approximately 65 km northeast of central Beijing."

L95: "attribute" -> "attributable" or "attributed"

Corrected.

Line 93:

"while cleaner conditions were attributed to dilution by clean air from the northwest."

L116-120: Here is just a repetition of the previous paragraph!

Apologies for the repetition. To clarify, we have thoroughly reviewed the manuscript to correct any editorial and grammatical errors, thereby enhancing the readability of our work.

L124-125: "As the optical diameter of BCc is largely independent of the morphology of BCc". This assumption may not be always accurate. Need clarification of the method to derive the optical diameter from the light-scattering signal.

We have added related discussions.

Line 117-122:

"As suggested by Moteki et al. (2010), for size smaller than 400nm, the scattering signal measured by SP2 is predominantly influence the volume of particle, with little influence from the morphology of BCc. The scattering at $\lambda=1064$nm which is larger than the particle size of a typical BCc, can minimize the geometric influence of scattering due to the complex morphology of BC. This is validated to be within 10% accuracy for the

optical size determination of BCc at 1064 nm (Hu et al., 2021). Thus, the measured optical diameter by SP2 is assumed to be equal to the volume equivalent diameter ($D_{ve}$)."

L143-144: "The kbc was test ..." I couldn't understand this sentence. Please check grammatical correctness.

This sentence was rewritten.

Line 143-144:

"The $\kappa_{BC}$ was tested to successfully predict the CCN activation properties of BCc in the field in-situ cloud observation."

L150: "governed" -> "controlled" or "operated"

Corrected.

Line 149-150:

"the DMA was controlled by custom-made software."

L160: "the other" -> "another"

Corrected.

Line 160:

"and another SP2 with a flowrate of 0.12 L/min,"

L179: "challenging" -> "unlikely"

Corrected.

Line 178-179:

"It is well recognized that the activation of BCc into CCN is more unlikely compared to BCf of the same diameter,"

L196: "ration" -> "ratio"

Corrected.

Line 202-203:

"$\frac{[toluene]}{[benzene]}$ is the ratio of measured toluene and benzene in the atmosphere."

L198: "test" -> "tested"

Corrected.

Line 204-205:

"The dataset of $t_{age}$ in this study has been successfully tested to determine the photochemical reaction or production rates of secondary OA,"

L204: Clarify the criteria of the air mass classification.

The criteria of the air mass classification was added.

Line 209-213:

"Ensemble calculations were performed with 3-hourly, 1°×1° GDAS horizontal and vertical reanalysis wind fields to retrieve 36-hour backward air mass trajectories to the target site. A clustering analysis of the air mass trajectories was conducted using a systematic clustering method (Draxier &Hess, 1998), and a total of 216 trajectories was calculated. The entire observation periods were classified into 4 clusters."

Figure2: Panels e-g lack the horizontal axis label.

The figure was revised.

[Figure]

L218: "less distances of transportation" -> "shorter transport distances"

Corrected.

Line 227:

"shorter transportation distances,"

L269: Remove "respectively"

Corrected.

Line 279-280:

"Figure 4 illustrates a representative example of size-resolved activation of all-particle and BCc at SS = 0.1% and 0.3%."

Figure 5: Increase the resolution or size of the lower panels in Figure 5. They are hard to see.

The figure was revised.

[Figure]

L365: I guess "Moteki et al. 2012" referred here should be Moteki et al. 2007 GRL instead.

Corrected.

Line 376-377:

"The aging scale of a few hours observed here is generally consistent with previously measured BC aging influenced by anthropogenic pollution, such as an aircraft measurement (Moteki et al., 2007)"

L400: "It was also found that the number activated fraction of BCc was more sensitive in the SS range from 0.1% to 0.2%, a typical range for fog and stratiform thin clouds formation." -> This statement needs some references.

References was added.

Line 411-413:

"It was also found that the number activated fraction of BCc was more sensitive in the SS range from 0.1% to 0.2% (Hu et al., 2021a), a typical range for fog and stratiform thin clouds formation."

Reference:

Draxier, R. R., & Hess, G. D.: An overview of the HYSPLIT4 modelling system for trajectories, dispersion, and deposition, Australian Meteorological Magazine, 47, 4, 295–308, 1998.

Hu, D., Liu, D., Kong, S., Zhao, D., Wu, Y., Li, S., Ding, S., Zheng, S., Cheng, Y., Hu, K., Deng, Z., Wu, Y., Tian, P., Liu, Q., Huang, M., and Ding, D.: Direct Quantification of Droplet Activation of Ambient Black Carbon Under Water Supersaturation, J. Geophys. Res. Atmos., 126, e2021JD034649, 2021a.

Hu, K., Liu, D., Tian, P., Wu, Y., Deng, Z., Wu, Y., Zhao, D., Li, R., Sheng, J., Huang, M., Ding, D., Li, W., Wang, Y., and Wu, Y.: Measurements of the Diversity of Shape and Mixing State for Ambient Black Carbon Particles, Geophys. Res. Lett., 48, e2021GL094522, 2021b.

Moteki, N., Kondo, Y., Miyazaki, Y., Takegawa, N., Komazaki, Y., Kurata, G., Shirai T., Blake, D. R., Miyakawa, T., Koike, M.: Evolution of mixing state of black carbon particles: aircraft measurements over the western pacific in march 2004. Geophysical Research Letters, 34(11), 235-255, 2007.

---

## Referee Report (RR1)

The author has revised it according to my suggestion, and I have no other question.